

# The impacts of climate change on the global range of *Culicoides punctatus* (Meigen, 1804) with notes on its status in Saudi Arabia

Esam S. Al-Malki

Majmaah University, Department of Biology, College of Science in Zulfi, Majmaah, Saudi Arabia

## ABSTRACT

Biting midges, particularly *Culicoides* species, pose significant health risks to humans and animals due to their biting behavior and ability to transmit diseases. Understanding their behavior and distribution patterns is crucial for predicting and controlling the spread of viral infections. This study employs species distribution modeling to assess the environmental suitability and potential future distribution of *Culicoides punctatus*, a species known for causing allergic reactions in horses and acting as a vector for bluetongue virus (BTV) and African horse sickness virus (AHSV). Species occurrence records for *C. punctatus* were collected from the Global Biodiversity Information Facility (GBIF), and environmental data representing climate variables were obtained from WorldClim. The data were used to develop species distribution models and predict the potential distribution of *C. punctatus* in the Old World under different emission scenarios. The results indicate that *C. punctatus* has a wide occurrence across the Old World, with the highest number of records in Europe. The species distribution models highlight the influence of climate on the distribution of *C. punctatus*, suggesting that climate change could impact its range and potentially expand regions with endemic viral infections. The study emphasizes the need for proactive measures to monitor and manage the spread of viral infections associated with *Culicoides* midges. The integration of geographic information systems (GIS) and remote sensing technology has facilitated high-throughput analysis techniques, eliminating the need for invasive experiments and enabling the remote assessment of species' habitats, land cover changes, and meteorology. Species distribution modeling, a powerful tool in ecological research, has been employed to predict the potential distribution of *C. punctatus* and assess its vulnerability to climate change. This study contributes to our understanding of the ecological implications of climate change on *Culicoides* midges and the associated viral infections. It provides valuable insights for designing effective management strategies, conservation efforts, and mitigation measures to minimize the impact of biting midges on human and animal health. Further research and monitoring are necessary to continuously update and refine these models in the face of changing environmental conditions.

Corresponding author
Esam S. Al-Malki,
e.almalki@mu.edu.sa

## INTRODUCTION

Biting midges, scientifically known as Ceratopogonidae, are a family of small, blood-feeding insects. Among the various genera within this family, one of the most notable is *Culicoides* (*Kettle, 1962*). *Culicoides* midges are known worldwide for their biting behavior, which can cause annoyance and discomfort to humans and transmit diseases to animals (*Parihar et al., 2022*). *Culicoides* midges are tiny insects, usually measuring 1–3 millimeters in length. They have slender bodies, long legs, and delicate wings (*Meiswinkel, Venter & Nevill, 2004*). These midges are typically active during the late afternoon and evening hours, preferring areas with high humidity and proximity to water sources like marshes, swamps, and wetlands (*Mullens et al., 2010*).

Female *Culicoides* midges require a blood meal to develop their eggs, and they are responsible for biting humans and animals (*Carpenter et al., 2013*). The bites can be painful, causing itching, swelling, and irritation. Some individuals may also experience allergic reactions to the bites, resulting in more severe symptoms (*Baker & Quinn, 1978*). One of the significant concerns associated with *Culicoides* midges, particularly certain species within the genus, is their ability to transmit diseases. The most notable disease *Culicoides* midges transmitted is bluetongue, which primarily affects ruminant animals like sheep, cattle, and goats (*Carpenter et al., 2008*). Bluetongue can have significant economic impacts on livestock industries due to decreased productivity and movement restrictions imposed to control the disease (*McDermott et al., 2015*). In recent years, another disease transmitted by *Culicoides* midges called Schmallenberg virus has emerged in Europe (*De Regge et al., 2012*). This virus also affects ruminant animals and can cause reproductive problems and congenital malformations (*Endalew et al., 2019*).

In Saudi Arabia, *Culicoides* midges are present and have been of concern due to their potential role as vectors of diseases (*Boorman, 1989*). Saudi Arabia is home to a diverse range of *Culicoides* species, and studies have identified numerous *Culicoides* midges in different regions of the country. The species composition may vary across various habitats and geographical areas within Saudi Arabia (*Alahmed, Kheir & Al-Khereiji, 2010*). *Culicoides* midges in Saudi Arabia have been implicated in transmitting several diseases. Bluetongue, caused by the Bluetongue virus, has been reported in livestock, including sheep, goats, and cattle (*Hilali et al., 2003*). Epizootic hemorrhagic disease (EHD), caused by the epizootic hemorrhagic disease virus, has also been detected in livestock in the country (*Istituto Zooprofilattico Sperimentale dell'Abruzzo e del Molise "G. Caporale", 2009*). These diseases can have significant impacts on animal health and productivity.

One of the *Culicoides* midges identified in Saudi Arabia is *Culicoides punctatus* (Meigen, 1804), (*Alahmed, Kheir & Al-Khereiji, 2010*) Like other *Culicoides* species, *C. punctatus* and its close relative *C. pulicaris* are known to cause allergic reactions in horses. They are also known to be vectors of bluetongue virus (BTV) (*Hilali et al., 2003*). AHSV can cause transient pyrexia of companion horses and donkeys in mild cases or impairment of the

respiratory system and subsequent effusions and hemorrhages in severe cases (*Carpenter et al., 2017*). AHSV, however, is not known to be endemic in Saudi Arabia, with only one case being diagnosed between 1959 and 1989 (*Mellor, Hamblin & Graham, 1990*), unlike the BTV, which is documented to have an alarmingly high seroprevalence rate in sheep, goats, cattle, and camels from different regions throughout Saudi Arabia (*Yousef, Al-Eesa & Al-Blowi, 2012*). Because both of these spread primarily through *Culicoides* spp., it is crucial to understand their behavior to enable prediction and control of the viral spread.

Climate change is expected to have an enormous effect on biodiversity worldwide, and *Culicoides* spp. is no exception to this. The United Nations has warned that the global average temperature may rise by up to 4 °C by the end of the 21$^{st}$ century relative to preindustrial values if drastic mitigation efforts are not undertaken (*Woodward et al., 2014*). Such a change will have disastrous environmental consequences, such as changes in precipitation patterns (*Dore, 2005*) more frequent extreme and dangerous weather conditions (*Hashim & Hashim, 2016*), and increased risk of natural disasters such as flooding and drought (*Berlemann & Steinhardt, 2017*).

In terms of biodiversity, climate change will endanger the survival of species that are less capable of adapting to quickly changing climatic conditions, potentially leading to their extinction (*Urban, 2015*). Some species or individuals may exhibit adaptability by modifying their typical behavior, morphology, or physiology (*Chown et al., 2010*), resulting in range shifting (*HilleRisLambers et al., 2013*), a phenomenon when a species adjusts its usual habitat range in response to environmental changes. Range shifting has the potential for a non-indigenous species to become invasive in its new habitat (*Moran & Alexander, 2014*). Climate change can result in ecological imbalances with significant environmental and economic repercussions (*Batten, 2018*). For species like *Culicoides* spp. that are known to transmit viruses, the movement of their habitats in response to climate change can be particularly risky. This is because it could expand places where viral infections are already present (*Elbers, Koenraadt & Meiswinkel, 2015*). An examination of the temperature preferences of three *Culicoides* species in the UK suggests that the geographical distribution and reproductive cycles of the studied species will expand in the near future due to climate change (*Wittmann & Baylis, 2000*).

The 21st century saw a significant increase in computing power and accessibility, leading to the advancement of geographic information systems (GIS). GIS refers to a computer system capable of visualizing, capturing, analyzing, and manipulating geospatial data (*Chang, 2017*). The combination of remote sensing technology with GIS is essential. This leads to the development of integrated GIS, which strives to enhance the transparency and compatibility between geographic information systems and remote sensing systems (*Hinton, 1996*).

The utilization of high-throughput GIS-based analysis methodologies allows for the examination of many Earth attributes through the analysis of data obtained from high-throughput remote sensing. This approach eliminates the necessity of conducting invasive tests in the field (*Sellers et al., 1990*). This not only results in significant cost savings but also allows remote deduction of the biophysical characteristics of the habitats of species (*Kerr & Ostrovsky, 2003*), monitoring changes in land cover and their impact on

biological systems (*Foody, 2008*), and conducting meteorology and climate studies using computational climate models (*Samanta et al., 2012*). Active remote sensing, a technique in remote sensing, enables the direct tracking of individuals' mobility. This method facilitates monitoring the behavior, migration, and dispersion of individuals in a species (*Salem, 2003*).

The availability of abundant environmental data and species occurrence data, together with the constant evolution of data analysis techniques, has created a tool called species distribution modeling (SDM). SDM algorithms use known species occurrence records and environmental variables to forecast the possible distribution of the species in other geographic regions (*Guisan & Zimmermann, 2000*). Based on the data provided to the SDM algorithm, which can include future projected ecological data, it can also be utilized to evaluate the future possible spread of a species. This tool has a wide range of environmental applications, including biogeography studies, conservation plan development, meteorological research, and effective management measures for pests and invasive species (*Elbers, Koenraadt & Meiswinkel, 2015*).

This study aims to employ species distribution modeling to assess the environmental suitability of *C. punctatus* in the Old World, infer the relationship between climate and biting midge-associated viral infections, and estimate its potential future distribution under climate change in different emission scenarios. This may help shed light on the future epidemiology of viral infections that spread *via C. punctatus* as a vector by estimating the future range of biting midges correlated with their associated viral infections.

## MATERIALS AND METHODS

### Data collection

#### Species occurrence records

The species occurrence data for *C. punctatus* were obtained from the Global Biodiversity Information Facility (GBIF) (https://doi.org/10.15468/DL.XZ36FW) (*GBIF.Org, 2023*), an open-access biodiversity database that combines information from published research and users who voluntarily contribute to the database with data from the literature (https://www.gbif.org/) (*Telenius, 2011*). After downloading the data in Darwin Core (DwC) format, the metadata was removed to extract only the latitude and longitude coordinates for authentic occurrence reports. This data was then stored in the comma-separated values (CSV) file format to ensure compatibility with the MaxEnt species distribution modeling tool. In addition, all of the data present in the literature were digitized and transformed into points of the same file format that could be used when building the model. This data was supplemented by several resources from across Saudi Arabia (Material S1). In this study, 1,217 occurrence reports in total were used (Fig. 1).

### Environmental data

The environmental data was provided by WorldClim version 1.4 and then transformed into 19 bioclimatic variables to reflect the climate: (near current: *WorldClim, 2024a*. Future: *WorldClim, 2024b*) (*Hijmans et al., 2005*). The future projections are derived from

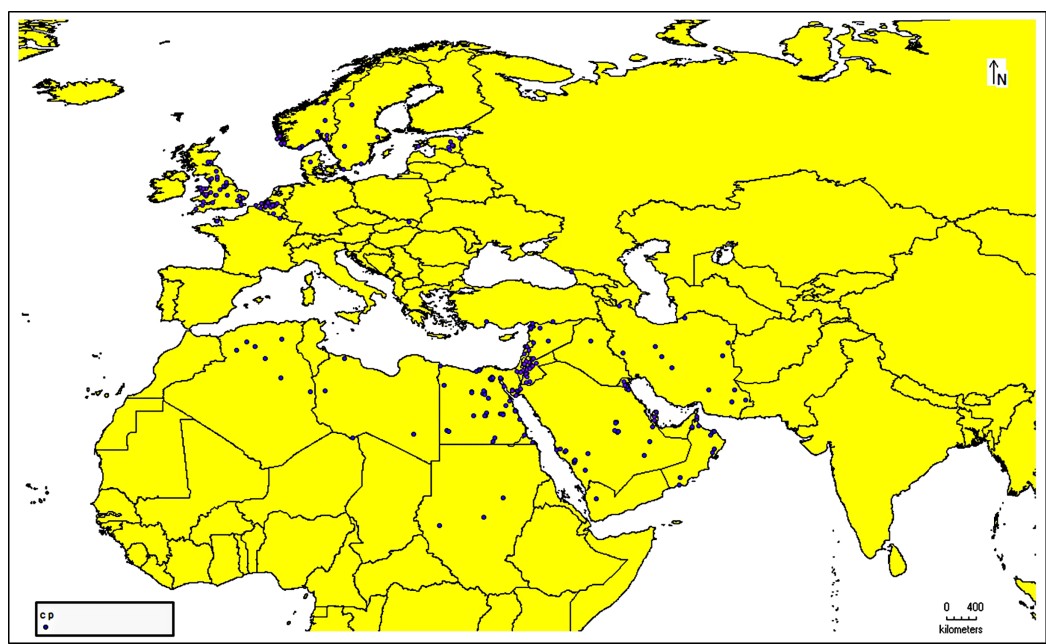

**Figure 1 Occurrence records map.** Cropped map of the Old World showing occurrence records of *Culicoides punctatus.*

various greenhouse gas emission scenarios. These variables display the monthly, seasonal, and annual averages, maximums, and minimums of precipitation and air temperature over different periods.

This data was downloaded at the equator at a spatial resolution of 2.5 min, or approximately 4.5 km (*Hijmans et al., 2005*). I used historical mean climate data from 1960 to 1990 to simulate the current climate scenario. For probable future climate change scenarios, the average values from 2041–2060 for 2050 and 2061–2080 for 2070 were used. The representative concentration paths (RCPs) used with the forecasted climatic data were RCP 2.6 and RCP 8.5, representing the best-case and worst-case scenarios, respectively (*van Vuuren et al., 2011*). The Japanese Meteorological Research Institute (MRI) created the MRI-CGCM3 general circulation model as the basis for future climate models (*Yukimoto et al., 2012*).

Thus, four datasets of future climate data—one for each of RCP 2.6 and RCP 8.5 in 2050 and 2070—were used in this study. Using ESRI ArcMap version 10.3, all climate data— historical and projected—was cropped only to include relevant Old World areas. It was then translated and exported into ASCII raster file format, allowing MaxEnt to use it as well.

## Model construction

Using the DIVA-GIS version 7.5 BIOCLIM technique, a preliminary overall model for the likely present distribution of *C. punctatus* was developed (*Sunil et al., 2009*). BIOCLIM modeling is utilized to provide first-pass geographical forecasts since it requires less computational power and resources (*Freile, Parra & Graham, 2010*). A more complex

machine-learning-based modeling approach, such as MaxEnt, built following BIOCLIM, can then be used to supplement and increase the accuracy of the model.

Following loading environmental parameters and species occurrence records, the DIVA-GIS BIOCLIM model generated distribution maps instantaneously. The species distribution models for *C. punctatus* were developed using maximum entropy species distribution modeling, which was used in the MaxEnt tool for primary models (*Phillips, Anderson & Schapire, 2006*). The maximum entropy technique was chosen because of its ability to generate a species distribution model using only species presence data, without the necessity for species absence data, and its good quality and reliability of results (*Phillips & Dudík, 2008*). The environmental variables and species occurrence records were imported into MaxEnt, just as they were in the BIOCLIM model. Each run done with MaxEnt used the following settings: The 10 replication runs used response curves, jackknife testing, cross-validation across runs, 75% random training samples, and 25% random testing samples. The output format was also changed to complementary log-log, or cloglog, better to estimate the likelihood of environmental suitability for *C. punctatus*. The cloglog output format is easy to comprehend because it is a binary scale, with 1 signifying the most suitable environments and 0 denoting the least suitable. The median of the 10 repetitions was chosen to represent the outcome of each run, which included 10 replicates in total. The maximum number of background points and the number of iterations were 10,000 and 1,000, respectively. To determine each climate variable's predictive potential and contribution, a prototype MaxEnt model was first built utilizing the previously specified species occurrence records and all 19 climate variables. When choosing significant environmental variables based on the removal of collinearity. The Pearson correlation coefficient was employed to assess the correlation between each pair of covariates to address collinearity among variables. Covariates with an absolute correlation coefficient ($r^2$) greater than or equal to 0.8 were considered highly correlated. The SDM Tools function in ESRI ArcGIS 10.7 was utilized to mitigate this collinearity. Specifically, the "Explore climate data" tool was used, which included a feature to remove highly correlated variables. By implementing this function, the correlation among covariates was effectively eliminated, ensuring the integrity of the analysis. Bioclimatic variables 8, 9, 18, and 19 (mean temperature of wettest quarter, mean temperature of driest quarter, precipitation of warmest quarter, and precipitation of coldest quarter, respectively) were avoided because they were known to cause spatial artifacts and discontinuities that would negatively affect the quality of the model (*Booth, 2022*).

The most significant environmental predictors were then identified using the jack-knife test, which is based on determining the significance of each variable by testing the difference in the model that occurs when said variable is left out (*Baldwin, 2009*). The most significant variables were then used for all subsequent modeling, while those deemed insignificant were excluded from the models.

The identical species occurrence records were imported into MaxEnt, but only with the environmental variables that were considered necessary in order to estimate the possible present range of *C. punctatus*. Four models were made in the same manner to represent the future possible distribution under various climate change scenarios: one for each of the

years 2050 and 2070 under the RCP 2.6 and RCP 8.5. The future climate data was fed into MaxEnt's predictions layer to enable future predictions based on the current observations.

## Quality assessment

The true skill statistic (TSS) and the area under the receiver operating characteristic (ROC) curve, or AUC, were employed to assess the performance of the model (*Phillips, Anderson & Schapire, 2006*; *Shabani, Kumar & Ahmadi, 2018*). The AUC is a threshold-independent technique for determining the predictive power of classification models such as MaxEnt. On a scale of 0 to 1, 0.9–1 indicates exceptional quality, 0.8–0.9 good quality, 0.7–0.8 satisfactory quality, 0.6–0.7 lousy quality, and less than 0.6 model failure (*Lissovsky & Dudov, 2021*).

To calculate the TSS, a threshold-dependent metric, the output model must be classified as positive or negative based on a threshold. The genuine favorable and actual negative rates are then combined and subtracted from one, providing a value between −1 and 1. Negative numbers represent a model whose performance is worse than random, positive values reflect a hypothetical perfect model capable of correctly distinguishing between positive and negative classes on every dataset, and zero indicates a model whose performance is no better than random (*Shabani, Kumar & Ahmadi, 2018*). The model was divided into positive and negative predictions using a threshold of 0.6, representing a 60% likelihood of environmental suitability.

## Model visualization

Using ESRI ArcMap 10.3, each MaxEnt model was inspected independently and exported as a picture (*Liu et al., 2021*). Each class was color-coded after being classified using Jenks natural breaks optimization across a range of values to aid visual comprehension. Next, the models were divided into presence-absence maps using the same cutoff as the true skill statistic calculation. To calibrate them, the value of the prospective current distribution was subtracted from that of the potential future distribution (*Zhang, Zhang & Tao, 2019*). As a result, the final maps contain zeros to indicate no range change, positive values to indicate range gain, and negative values to indicate range loss. The maps were subsequently categorized, color-coded, and exported to make it easier to visually grasp the future potential distribution in relation to the existing potential distribution.

# RESULTS

## Model performance

The mean AUC value across the replicated MaxEnt runs was revealed to be 0.925, indicating that the model was of excellent quality, as shown in Fig. 2. The TSS value also indicates that the model performance is high, with a 0.7 value.

## Effects of environmental variables

Analysis of the jack-knife test, shown in Fig. 3, revealed that the most significant environmental variables for *Culicoides punctatus* were bio_4 (temperature seasonality), bio_6 (minimum temperature of coldest month), bio_11 (mean temperature of coldest

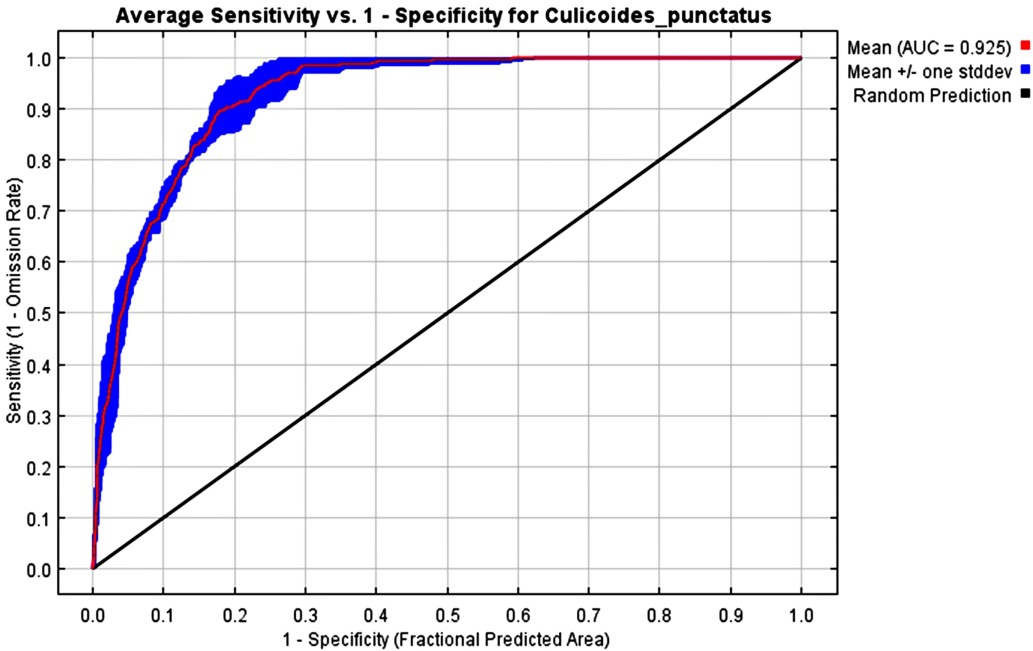

**Figure 2 AUC.** Receiver operating characteristics (ROC) curve of the MaxEnt model.

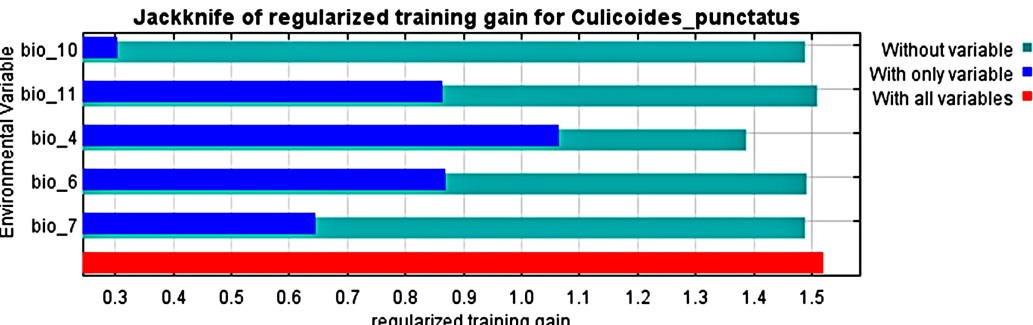

**Figure 3 Jack-knife test.** Jack-knife test of the most significant environmental variables.

quarter), bio_7 (temperature annual range), and bio_10 (mean temperature of warmest quarter) in descending order of significance.

The response curves revealed the most suitable ranges within these variables. By interpretation of bio_6, bio_10, and bio_11 (Fig. 4), it is strongly suggested that *C. punctatus* prefers environments that range from moderately warm climates to hot climates.

It is also worth noting that none of the precipitation bioclimatic variables were found to be significant in predicting the distribution of *C. punctatus*, potentially suggesting that it can live across a wide geographical range of environments regardless of the precipitation conditions throughout said environments and that its distribution is dependent mainly on seasonal temperature conditions.

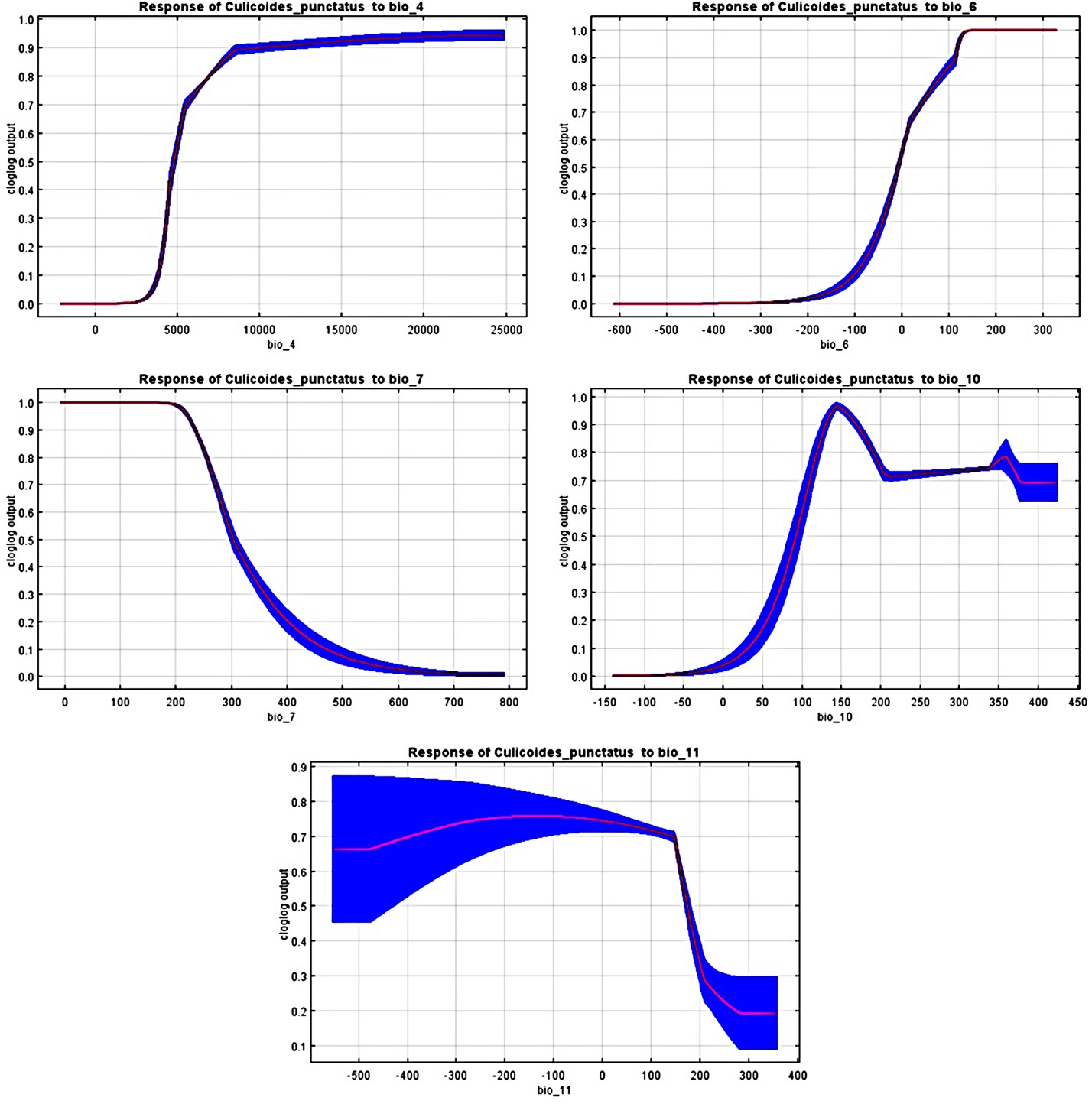

**Figure 4 Response curves.** Response curves of *C. punctatus* to significant environmental factors.

## Distribution maps

### Potential current distribution

The preliminary BIOCLIM model constructed for the distribution of *C. punctatus* under the historical climate scenario shows a peculiar range of suitable habitats in the Old World. As seen on the map, the environments with the highest suitability for *C. punctatus* appear to be the warm humid coasts of the Mediterranean, both in North Africa and in South Europe. Significant, highly suitable environments are also seen in the Sinai Peninsula and northwestern Saudi Arabia near the Jordanian border (Fig. 5).

The MaxEnt model of the *C. punctatus* distribution under the historical climate scenario revealed regions largely consistent with those shown in the BIOCLIM model, albeit with a larger area of suitable regions. Much like the BIOCLIM model, the MaxEnt model also shows the highest suitability areas for *C. punctatus* to be predominantly concentrated along the Mediterranean coasts, although the suitable regions in notable areas are greatly expanded in the area. These expansions include most of the Arabian Peninsula, the Nile Delta in Egypt, and almost the entirety of West Europe (Figs. 5 and 6).

In Saudi Arabia, the BIOCLIM model indicates that most of the Kingdome has low and medium suitability for *C. punctatus*, and only the northern west has high to very high habitat suitability. On the other hand, the Maxent modeling shows that the whole kingdom has high to very high habitat suitability, especially on the coast of the Gulf.

### Potential future distribution

The future projections of the MaxEnt models revealed that *C. punctatus* is predicted to still enjoy considerable range in the Old World (Figs. 7 and 8). At first glance, the models built on the increasingly severe climate change scenarios appear to display more severe range shifts, albeit while still enjoying considerable range. To simplify the interpretation and quantify the regions exhibiting range gains and losses, the calibrated maps will be used instead of the previous uncalibrated maps for future climate change scenarios.

The calibration maps in (Figs. 9 and 10) show alarming range shifts for *C. punctatus* under the future projected climate change scenarios used in this study. Given the rising average global temperatures, an exceedingly severe range gain is predicted for *C. punctatus* in most of the Old World. Specifically, there is an alarmingly high range gain in much of North Europe, Central Asia, the Indian Subcontinent, and many regions of North Africa and Sub-Saharan Africa.

In the most severe climate change scenario seen in 2070 under RCP 8.5 (Fig. 10B), there is also an observable range loss, albeit at a much more modest scale than the range gain, seen in some regions in central Saudi Arabia extending as far northeast into Iraq. For the most part, most of the range of *C. punctatus* appears to be projected to grow significantly in area, and a minority of its range seems to be relatively unaffected. This is in line with the interpretation of the response curves (Fig. 4), which show that *C. punctatus* prefers moderately warm to hot climates, and thus, a rising mean global temperature in the future under climate change may serve to provide more suitable habitats for *C. punctatus*.

It is also worth noting that with increasingly severe climate change scenarios, the extent of range shifting projected to be exhibited by *C. punctatus* also becomes increasingly more

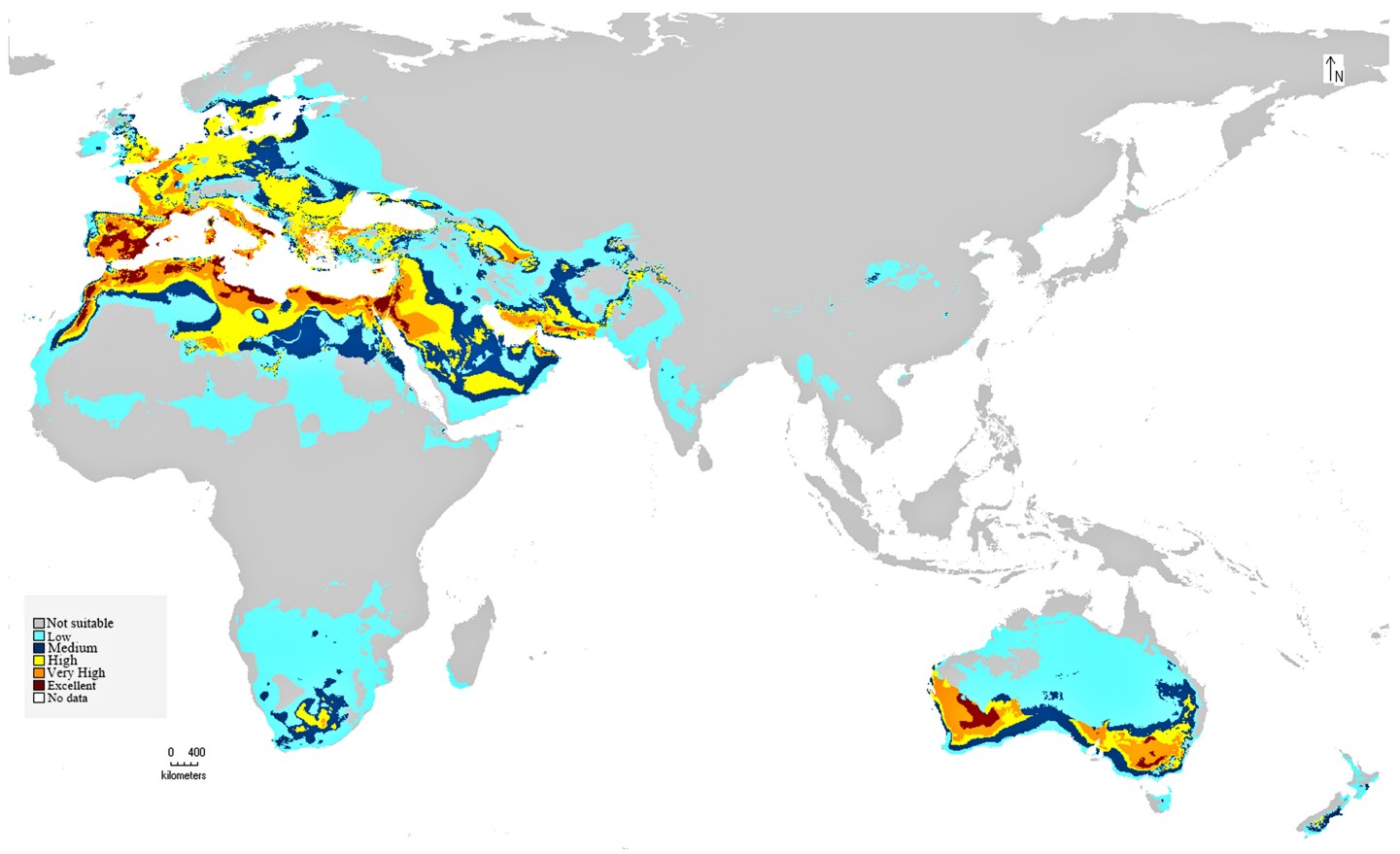

**Figure 5 BIOCLIM model for *C. punctatus* distribution.** Map of the BIOCLIM model for the distribution of *C. punctatus* under the historical climate scenario.

severe, such as when comparing (Fig. 9B) against (Fig. 9A), or likewise by comparing Fig. 10B against (Fig. 10A). In Saudia Arabia, the status of *C. punctatus* will not be significantly affected through all scenarios except for the loss of some suitable areas in the kingdom's central region in 2070 under RCP 8.5 (Fig. 10B).

## DISCUSSION

Biting midges are known to inhabit worldwide environments and are known disease vectors for various animals (*Meiswinkel, Venter & Nevill, 2004*). BTV, one of the diseases caused by *Culicoides* spp. of biting midges, is geographically limited exclusively to areas inhabited by *Culicoides* spp. and is widespread globally, including in the Old World, most notably in Africa and the Arabian Peninsula (*Tabachnick, 2004*). In one study, it was found that an epidemic of BTV directly caused the death of 1.5 million sheep in Europe in the few years following an outbreak, causing massive economic losses that were further amplified by the implementation of trade restrictions intended to prevent the spread of the virus as well as the development of secondary diseases in animals with weakened immune systems (*Ganter, 2014*).

Much like the BTV, the Schmallenberg virus is also known to be transmitted *via Culicoides* spp. as its vector (*Endalew et al., 2019*). One study conducted in Austria found

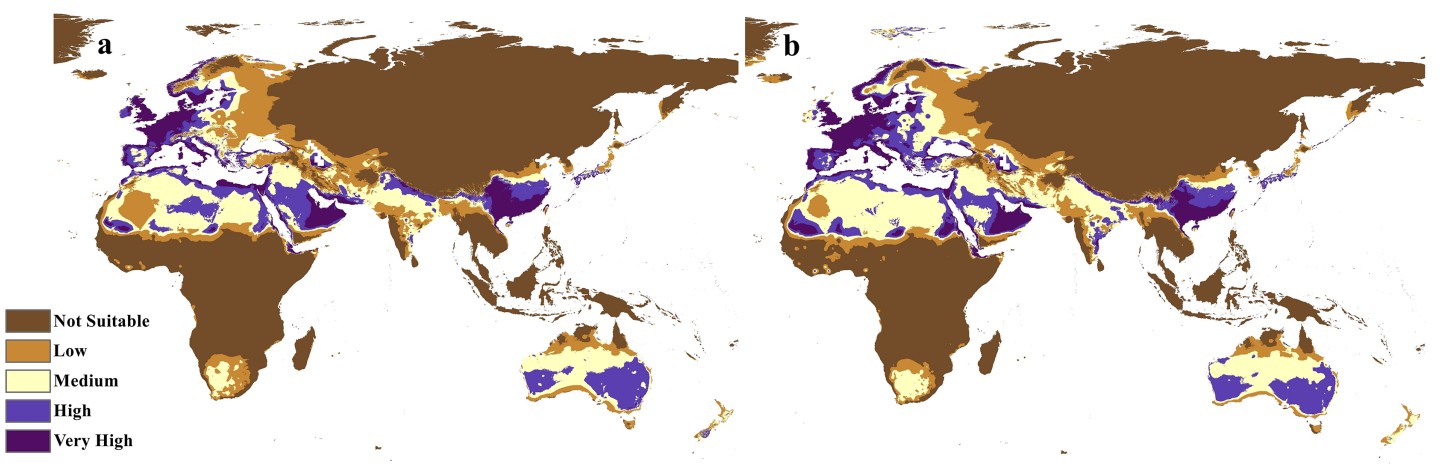

**Figure 6 Maxent Model distribution of *C. punctatus*.** Map of the MaxEnt distribution of *C. punctatus* under the historical climate scenario.

**Figure 7 Future distribution prediction for 2050.** Maps of the projected future distribution of *C. punctatus* in 2050 under (A) RCP 2.6 and (B) RCP 8.5.

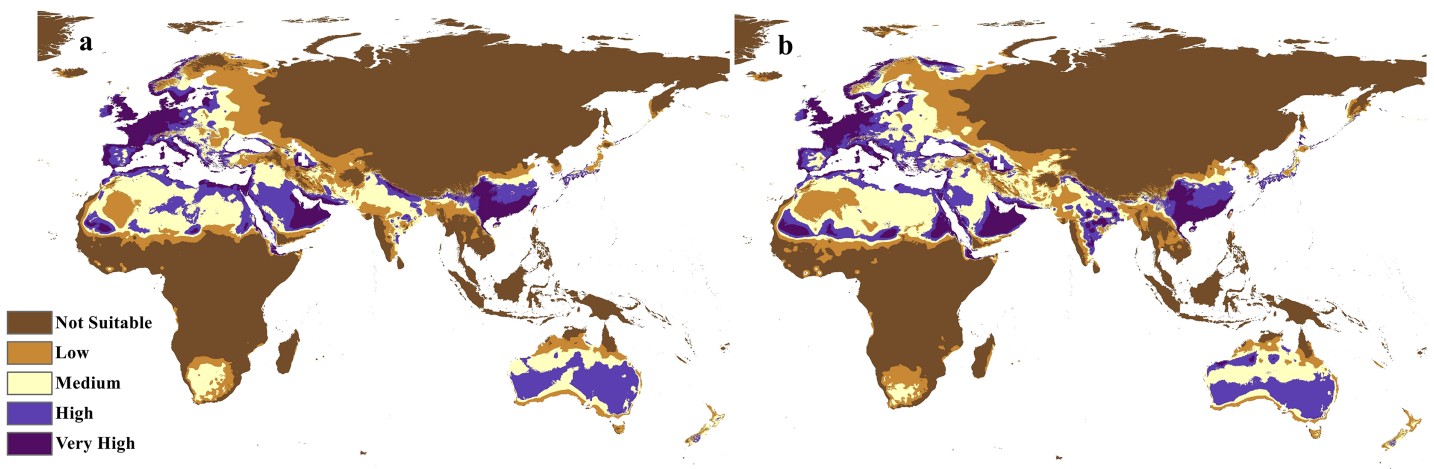

**Figure 8 Future distribution prediction for 2070.** Maps of the projected future distribution of *C. punctatus* in 2070 under (A) RCP 2.6 and (B) RCP 8.5.

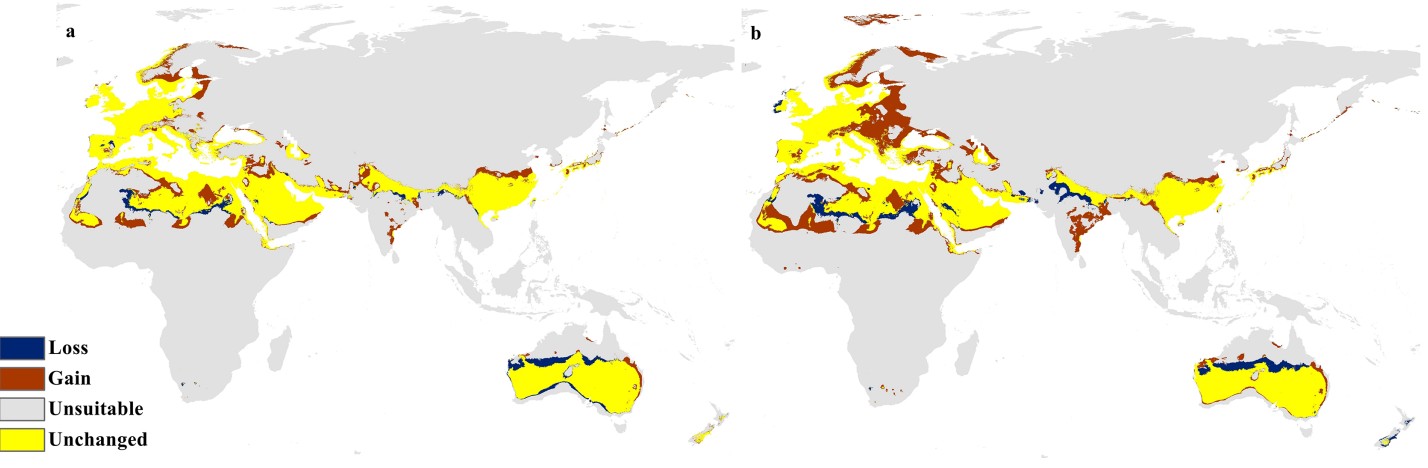

**Figure 9 Calibration maps for 2050.** Calibration maps of the projected distribution of *C. punctatus* in 2050 under (A) RCP 2.6 and (B) RCP 8.5.

that the Schmallenberg virus was associated with increased rates of ruminant spontaneous abortions and fetal death in Austria during the 2012 and 2013 Schmallenberg virus epidemic, as the genetic material of the Schmallenberg virus was found in the fetal tissues of aborted and newborn cattle as well as in their amniotic fluid (*Steinrigl et al., 2014*).

As with all other living organisms, *Culicoides* spp. are also subjected to the impacts directly or indirectly associated with climate change, including adaptation or range shifting (*HilleRisLambers et al., 2013*). This poses an inherent risk because it is a viral vector and already exhibits a relatively large global range. It is also increasingly concerning that a study insinuated that *Culicoides* spp. appear to prefer warmer temperatures and that their range may be increased in the future if rapid mitigation efforts are not undertaken to prevent further climate change (*Wittmann & Baylis, 2000*).
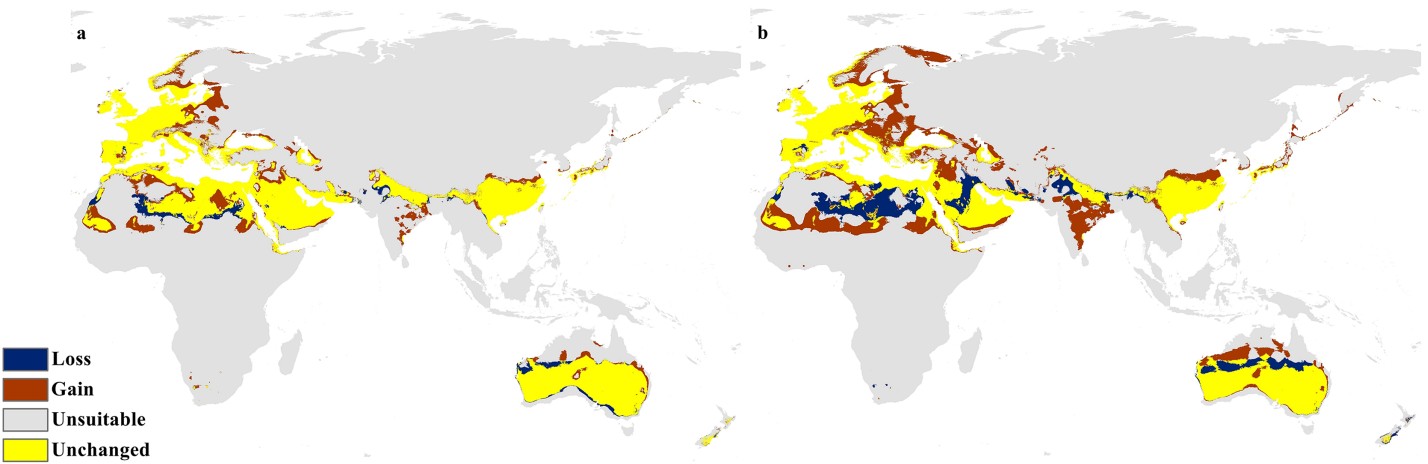

**Loss**
**Gain**
**Unsuitable**
**Unchanged**

**Figure 10 Calibration maps for 2070.** Calibration maps of the projected distribution of *C. punctatus* in 2070 under (A) RCP 2.6 and (B) RCP 8.5.

For these reasons, this study aimed to exemplify and identify the regions that are most suitable for a species of *Culicoides* biting midges, particularly *Culicoides punctatus*, in the Old World, as well as predict their future distribution under various climate change scenarios. This would potentially yield maps that can be used to define regions at the most risk of infections associated with *C. punctatus* so that necessary measures may be taken.

To this end, we employed maximum entropy modeling as implemented in the MaxEnt application (*Phillips, Anderson & Schapire, 2006*) using species occurrence records of *C. punctatus* that are available and open-source (*GBIF.Org, 2023*) along with bioclimatic variables representing both the current and future projected climate change scenarios (*Hijmans et al., 2005*). The generated models would then show the potential range of *C. punctatus* in the present and expected range shifts in the future. The performance of the models was judged according to their AUC and TSS values, where judgment according to the AUC revealed the model was of excellent quality (Fig. 2) (*Lissovsky & Dudov, 2021*) and assessment according to the TSS value revealed (0.7). The bioclimatic variables found to be most effective in predicting the range of *C. punctatus* were shown to be temperature seasonality, minimum temperature of the coldest month, mean temperature of the coldest quarter, temperature annual range, and mean temperature of the warmest quarter, in descending order of predictive important (Fig. 3). The response curves (Fig. 4) also showed that *C. punctatus* prefers moderately warm to hot climates, as was insinuated by a previous study (*Wittmann & Baylis, 2000*). Various species distribution modeling approaches have been previously employed to model other *Culicoides* spp., including a study on *C. imicola* Kieffer, 1913 in Spain (*Peters et al., 2014*).

Maps of the generated species distribution models (Figs. 6 through 10) reveal large suitable ranges for *C. punctatus*, which inherently represents a risk to animal husbandry and a potential economic risk to the food supply. This is especially concerning in regions where the economy is already dependent on mixed agriculture, as is the case along the Nile Delta in Egypt and the northern areas of Sudan (Fig. 6–10).

The future calibration maps (Figs. 9 and 10) show substantial range shifts in the distribution of *C. punctatus* in the different climate change scenarios being studied. The most alarming regions are in North Europe, Central Asia, the Indian Subcontinent, and much of North Africa and Sub-Saharan Africa. This is in line with the observation that *C. punctatus* prefers moderately warm to hot climates, as the projected increases in global mean temperatures will directly cause an increase in suitable ranges for *C. punctatus*. The predicted northbound shift in the distribution of *C. punctatus* in Europe due to the climate change appears to be consistent with that of other *Culicoides* species; for instance, *Culicoides sonorensis* was also projected *via* species distribution modeling to shift northward as the Earth warms (*Zuliani et al., 2015*).

It is also worth noting that the BIOCLIM model in Fig. 5 appears to exhibit a much smaller range than the MaxEnt model in Fig. 6, which is possibly explained by the fact that newer species distribution modeling algorithms such as MaxEnt can perform better and with higher sensitivity with relatively few environmental variables, relative to an older algorithm such as BIOCLIM (*Booth, 2018*).

Our results show an urgent need to take preventive measures against *Culicoides punctatus* and potentially other related *Culicoides* spp. Vaccinations against bluetongue (*Roy, Boyce & Noad, 2009*) and Schmallenberg virus (*Wernike et al., 2013*) are currently being researched. However, further evaluation of these studies is needed to assess their efficacy as well as the duration of immunity provided. Nevertheless, developing a successful vaccine would significantly mitigate the risks associated with *Culicoides* spp.

However, in addition to developing approaches to mitigate the risks associated with Culicoides spp., we also want to urgently note the impacts of climate change on diseases that are not exclusively limited to natural disasters (*Berlemann & Steinhardt, 2017*), contrary to the common misconception. Climate change is expected to hasten the spread of some infectious diseases and impede that of others (*Shope, 1991*). This can be primarily attributed to changes in ecological balances, which themselves result from climate change-induced range shifting.

Notably, the species distribution models for *C. punctatus* in this study were constructed using only bioclimatic variables, without accounting for species-species interactions or other biotic variables that may affect their distribution. This may provide a satisfactory estimation of habitat suitability for *C. punctatus*, but it may be further improved by accounting for additional biotic factors or by conducting smaller local and/or regional studies on a higher resolution.

It should also be noted, additionally, that MaxEnt is a machine learning algorithm that is subject to error and produces models that may be different from other similar algorithms, such as random forests, artificial neural networks, and support vector machines. In particular, MaxEnt provides an advantage over many other models when environmental variables tend to be highly correlated, but may be outperformed by deep neural networks on very large datasets with high complexity (*Zhang & Li, 2017*; *Srivastava, Lafond & Griess, 2019*). Given the scale of this study, along with the number of environmental variables used in the model, we believe MaxEnt provides a satisfactory estimation.

## CONCLUSIONS

In conclusion, this study highlights the importance of understanding the behavior and distribution patterns of *Culicoides* midges, particularly *C. punctatus*, in the context of predicting and controlling the spread of viral infections. The research utilized species distribution modeling to assess the environmental suitability and potential future distribution of *C. punctatus* in the Old World under different emission scenarios. The findings indicate that *C. punctatus* has a wide occurrence across the Old World, with Europe having the highest number of records. Climate was identified as a significant factor influencing the distribution of *C. punctatus*, suggesting that climate change could impact its range and potentially expand regions with endemic viral infections. This emphasizes the need for proactive measures to monitor and manage the spread of viral infections associated with *Culicoides* midges. Integrating geographic information systems (GIS) and remote sensing technology has played a crucial role in enabling high-throughput analysis techniques, eliminating invasive experiments, and allowing for remote assessment of species' habitats, land cover changes, and meteorology. As a powerful tool in ecological research, species distribution modeling has provided valuable insights into the potential distribution and vulnerability of *C. punctatus* to climate change. The implications of climate change on *Culicoides* midges and the associated viral infections are significant, as the shifting ranges of these vectors may lead to the expansion of regions with endemic infections. This poses ecological and economic consequences, particularly in the livestock industry. Therefore, effective management strategies, conservation efforts, and mitigation measures should be designed based on the understanding provided by this study. Further research and monitoring are necessary to continuously update and refine the models in response to changing environmental conditions. This will contribute to an improved understanding of the ecological implications of climate change on *Culicoides* midges and provide a basis for designing targeted interventions to minimize the impact of biting midges on human and animal health. Overall, this study emphasizes the importance of considering the ecological dynamics of *Culicoides* midges and their interactions with climate change in order to develop comprehensive strategies for disease control and mitigation in the face of global environmental challenges.

## ACKNOWLEDGEMENTS

The author expresses sincere gratitude for the use of the facilities at the College of Science in Zulfi, Majmaah University, which greatly contributed to the completion of this manuscript. Also, I would like to thank Marin Redwan from USAID for the linguistics revision of the manuscript. Thanks also go to Prof. Mohamed Nasser from Research lab of Biogeography and Wildlife Parasitology, Department of Entomology, Faculty of Science, Ain Shams University for his scientific approval of the manuscript.

### Funding

The author received no funding for this work.

## Competing Interests

The author declares that they have no competing interests.

## Author Contributions

- Esam S. Al-Malki conceived and designed the experiments, performed the experiments, analyzed the data, prepared figures and/or tables, authored or reviewed drafts of the article, and approved the final draft.

## Data Availability

The data is available at:

- https://www.worldclim.org/data/v1.4/worldclim14.html.
- https://www.worldclim.org/data/v1.4/cmip5_2.5m.html.

## Supplemental Information

Supplemental information for this article can be found online at http://dx.doi.org/10.7717/peerj.18916#supplemental-information.

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
