# Peer review of "The impacts of climate change on the global range of Culicoides punctatus (Meigen, 1804) with notes on its status in Saudi Arabia"

_PeerJ, doi:10.7717/peerj.18916_

## Round 0.1 · original submission · Major Revisions

Dear Dr. Al-Malki,

after this first review round, both reviewers provided a number of suggestions that qualify this review round as a major review. Along with text improvements, the names of the variables should be changed, and a better geographic representation of the study should be made. Also, more detailed improvements suggested by reviewer #2 need to be implemented in the new version of your manuscript.

Reviewer 2 has suggested that you cite specific references. You are welcome to add it/them if you believe they are relevant. However, you are not required to include these citations, and if you do not include them, this will not influence my decision.

Sincerely,
Daniel Silva

·

Basic reporting

Overall, language is adequately used through the paper. The exception is on scientific names of the species, which are very often not italicized. There is enough background provided, both on study objects and methods applied. The only background that may be lacking is regarding assumptions and limitations of SDMs, that I feel should be discussed and presented more completely on the study.

Figures and tables are adequate. In some cases, like in figures 2 and 3, I recommend using the complete variable names (q.g. mean temperature), not their abbreviations (bio_1, for example). I also suggest the addition of another figure, displaying the occurrence points used in the study.

Results presented are adequate answers to the studies’ questions. A clear prediction on the possible range changes of the species is presented.

Experimental design

It is a primary study within aims of the journal. A quick survey in the literature suggests is it an original paper. The methods are adequate for an important and well-defined question.

Methods are conducted adequately, with the exception of dealing with collinearity on environmental data, which is a topic I presented directly to the author in more details. Methods are overall described with enough detail; I’ve sent suggestions to the authors on which additional details are needed.

Validity of the findings

Results are valid, although a better treatment on environmental variables collinearity would be very beneficial. Worldclim and GBIF data are extensively used in literature and are overall trusted. The occurrence dataset was provided, and a quick inspection suggests it is consistent.

Additional comments

This is a consistent study, that uses an extensively studied method (species distribution modelling) for predictions on an economically important insect species. Though I have negative opinions on some choices, e.g. the usage of only two model algorithms (Maxent and Bioclim), the inadequacy of exclusion of environmental variables and the lack of a method for treating collinearity among environmental variables, I am aware those choices are often performed in similar studies. Those are suggestions I would like the author to know, and I also would like to have more specialists’ opinions on the subject.

I have some suggestions to the author, that I believe would make the paper better: 1) the addition of a geographical presentation of the occurrence data; 2) The incorporation of a methodological step in order to deal with environmental collinearity; 3) A deeper discussion and presentation of niche models’ shortcomings and limitations, just as a little more detail on some choices and procedures (e.g. scale choice, duplicate removal, etc); 4) there are some more scattered suggestions throughout the text, which will be sent to the author via another document.

Because the study is adequate, although I have some important suggestions to the authors (both in the sense of additional information/presentation and maybe some needed corrections), I would suggest the paper should be accepted with at least minor suggestions and corrections.

Reviewer 2 ·

Basic reporting

In the manuscript “The impacts of climate change on the global range of Culicoides punctatus (Meigen, 1804) with notes on its status in Saudi Arabia” the authors used species distribution modeling to assess the environmental suitability and potential future distribution of Culicoides punctatus, a species known for causing allergic reactions in horses and acting as a vector for bluetongue virus (BTV) and African horse sickness virus (AHSV). This manuscript is well organized, and the drawn conclusions are coherent with the obtained results.

Lines 42 – 43: Please, arrange the keywords alphabetically.

Lines 46 – 85: Please, reduce this part of the manuscript.

Lines 97 – 98: I think that you should add these important references as examples to support your sentence: “Range shifting may pose a risk for a non-indigenous species becoming invasive in its new environment.”. I would like to suggest:

Di Febbraro, M., et al., (2023). Different facets of the same niche: Integrating citizen science and scientific survey data to predict biological invasion risk under multiple global change drivers. Global Change Biology, 29(19), 5509-5523.

Song, R., et al., (2023). A temporal assessment of risk of non-indigenous species introduction by ballast water to Canadian coastal waters based on environmental similarity. Biological Invasions, 25(6), 1991-2005.

Lines 101 – 111: Please, reduce this part of the manuscript.

Lines 112 – 120: Please, delete this sentence.

Lines 127 – 128: I think that you should add these important references as examples to support your sentence: “ranging from biogeography studies, conservation strategies and planning thereof, meteorology research”. I would like to suggest:

Buonincontri, M. P., et al., (2023). Shedding light on the effects of climate and anthropogenic pressures on the disappearance of Fagus sylvatica in the Italian lowlands: evidence from archaeo-anthracology and spatial analyses. Science of The Total Environment, 877, 162893.

Guiquan, S., et al., (2023). Geographic distribution and impacts of climate change on the suitable habitats of Rhamnus utilis Decne in China. BMC Plant Biology, 23(1), 592.

Lines 130 – 133: Please, explain in detail you hypothesis and predictions.

Experimental design

Lines 138 – 148: Have you analyse the occurrence distribution for spatial autocorrelation by using some statistical methods such as Moran I?

Lines 177 – 180: I think that you should these important references as examples to support your sentence: “The choice of the maximum entropy algorithm was based on its capacity to create a species distribution model based just on species presence data, without the need for species absence data, and on its excellent quality and reliable findings.[46].”. I would like to suggest:

Bosso, L., et al., (2023). Integrating citizen science and spatial ecology to inform management and conservation of the Italian seahorses. Ecological Informatics, 79, 102402.

Li, X., et al., (2023). Mapping cropland suitability in China using optimized MaxEnt model. Field Crops Research, 302, 109064.

Zhang, Y. F., et al., (2023). Prediction of global potential suitable habitats of Nicotiana alata Link et Otto based on MaxEnt model. Scientific Reports, 13(1), 4851.

Lines 181 – 208 : Please, add all the details for each setting used in in maxent models.
Lines 210 – 211: I think that you should these important references as examples to support your sentence: “The true skill statistic (TSS) and the area under the receiver operating characteristic (ROC) curve, or AUC, were used to evaluate the quality of the model [45].[49].”. I would like to suggest:

Salinas-Ramos, V. B., et al., (2021). Artificial illumination influences niche segregation in bats. Environmental Pollution, 284, 117187.

Brown, C. H., & Griscom, H. P. (2022). Differentiating between distribution and suitable habitat in ecological niche models: A red spruce (Picea rubens) case study. Ecological Modelling, 472, 110102.

Wang, Y., et al., (2023). Modeling seasonal changes in the habitat suitability of Coilia nasus in the Yangtze River Estuary using tree-based methods. Regional Studies in Marine Science, 67, 103212.

Validity of the findings

Line 238 : Please, delete all the references from the results section. You CANNOT add refernces in the results.

Lines 304 – 377 : Discuss your results also with those already abtained from other studies on different organisms.

Additional comments

Figures 1 and 2 : To move in the supplementary materials.
Figures 4 – 9 : Please, add the north symbol and the scale in the maps.

---

## Round 0.2 · Minor Revisions

Dear Dr. Al-Malki,

After this new review round, both reviewers reached more positive decisions regarding your manuscript in comparison to your last version. Still, text "re-usage" and issues with the italic names of the species were found. I believe the use of a third-party company for editing and proofreading your manuscript may be of hand, and there are several companies available in the market for that purpose. Please take care of these issues while preparing the next version of your manuscript. As soon as the final reviewer believes your manuscript is worthy of publication, it will certainly be accepted for publication.

Sincerely,
Daniel Silva

**Language Note:** The Academic Editor has identified that the English language must be improved. PeerJ can provide language editing services - please contact us at [email protected] for pricing (be sure to provide your manuscript number and title). Alternatively, you should make your own arrangements to improve the language quality and provide details in your response letter. – PeerJ Staff

·

Basic reporting

I believe the paper is well presneted, as methodological decisions taken are explicit, and the work is viable for publication. I came to see that the other reviewer found some levels of re-usage for text, and I also believe the paper may only be published If this issue is really addressed.

Experimental design

The authors have taken more actions onto dealing with collinearity in environmental data, following my suggestions from the first reviewing process. Personally, I don’t know much about the ArcGis tool used, and I would suggest, optionally, a more comprehensive presentation on what it exactly performs to the data. However, I don’t see this as point as an impediment to publication. Authors have included a map with occurrence points, and I think the methods are now adequately presented for publication.

Validity of the findings

I do not wish to add any comment on this section, beyond what i already did in the first review.

Additional comments

- There is still a large number of scientific names not italic, especially in the discussion section. It is mandatory to correct this.

- In the introduction (lines 106 and 107) the authors present reasoning, with references, in favor of the “complete elimination for the need for field-based invasive experiments”. I believe this is far from the truth. All models are simplifications from reality, they all have unrealistic assumptions and can make mistakes. To cite some famous phrasing, “all models are wrong, but some are useful”. Therefore, modelling helps to achieve some advances without the need for field studies (although the occurrence data used by authors probably come from field studies), but I strongly disagree from a complete elimination for the need of field studies. I would like to see this phrasing addressed, but I also understand that this is optional.

- In line 249, in the methods sections, authors argue that precipitation is not important for the species distribution, and this information seems contradictory to the preference for humid habitat presented in the introduction section. I’m aware there is a big difference on scale on how those two informations are provided. But I still think It would be beneficial if the authors assumed in the methodology section that this may be an artefact generated by the model, by collinearity or other factors. This is, however, optional to the authors.

- In the discussion section (line 359) authors argue that Maxent produced a larger range area because it is a better model. This is not true. The better model is not the one which produces larger ranges, it is the one which produces ranges more similar to what are the actual ranges of the species in question. I would suggest the authors to present that Maxent generated a larger range, and exclude the justification about being a better model. Maxent is presented in the scientific literature as a powerful modelling tool, but we cannot assume it will always generate better predictions than other models.

Reviewer 2 ·

Basic reporting

Well done!

Experimental design

Well done!

Validity of the findings

Well done!

Additional comments

Well done!

---

## Round 0.3 · Major Revisions

Dear Dr. Al-Malki,

After the review round I made, I found that many of the suggestions required by the reviewers were not completed. In addition, I found a plentitude of issues that still hinder the acceptance of your manuscript for publication in PeerJ in its current form. I hope they help you to improve the manuscript presentation for your next submission in PeerJ. In addtion, please consider paying for a third-party company to edit and proofread your manuscript before future submissions.

Sincerely,
Daniel Silva

L28: Missing italics in C. punctatus in the abstract.
L33: Please avoid possessives in academic texts. It is too coloquial.
L42-43: Please separate the Keywords from your abstract.
L50: Please use en dashes in numerical ranges.
L74-75: Please invert the acronyms and their explanations. THe explanation must come outside the parenthesis and the acronym inside it. "bluetongue vírus (BTV) in addition to African horse sickness virus (AHSV)"
L93: There is a sentence here that makes no sense "The user's text is "[25]"" Please correct it.
L94: The citation [26] is after the period sign.
L97 and 102: Please include a space after "[27]." and "[29]."
L110: Please avoid possessives in academic texts. It is too colloquial. Citations 32 and 33 are after the commas signs. They need to be before them.
L133: Please provide the doi number from GBIF regarding the data you used.
L139: Please avoid possessives in academic texts. It is too coloquial.
L141: "Figure 1 needs to be inside parenthesis.
L146: There is a missing period after "[40]"
L150: Citation [40] needs to come before the period sign.
L156: The same for citation [42]
L154: Citation [41] needs to come before the period sign.
L160: THe "model construction" subheading is missplaced.
L162: C. punctatus needs to be in italics. Citation [43] is after the period sign.
L165: Please avoid possessives in academic texts.
L173: Citation [46] is after the period sign.
L175: "10" instead of "ten" here
L178: Please avoid possessives.
L182: "1,000"
L195: Please avoid possessives.
L210: Please avoid possessives. Citation [49] needs to be before the period sign. You need to cite Allouche et al. 2006 here, instead of citation 49.
L212 and 213: Please use en dashes in numerical ranges.
L213: Regarding TSS, if you read Allouche et al. 2006, you will see that a 0.5 value is acceptable.
L216-217: Please use the minus symbol (ALT key + 8722 in Windows) instead of hyphen.
L220: Citation [49] needs to be before the period sign.
L221: There are two period signs at the end of the sentence.
L224: Citation [51] needs to be before the period sign.
L228: Citation [52] needs to be before the period sign.
L245: Please separate "figure4"
L267: Please use "(Figures 5 and 6)".
L273-274: Please avoid one sentence paragraphs. "(Figures 7 and 8)"
L275-279: This seems to be a methodological decision. It needs to come in the M&M section. Not here!
L280: "Figures 9 and 10"
L285: "(Figure 10B)" Please note that Figure 10A was not called out in the text.
L294: C. punctatus is missing the italics.
L290: "Figure 4"
When you call figures, there is no need to use hyphens to separate their numbers and letters. The correct spelling is "Figure 10b," not "Figure 10-b."
L293-296: Please avoid one sentence paragraphs. Please use italics in the name of the species. Please use "Figure 9A and 9B" and "Figure 10B against Figure 10A"
L299: "Figure 10B"
Figure 1: Please improve this figure. There are many areas in the east without any occurrences that need to be cut. This also occurs to the South of the figure.
Figure 2: The legend is duplicated. This figure needs to be a supplementary material.
Figure 3: The legend is duplicated.
Figure 4: The legend is duplicated.
Figure 5: C. punctatus is missing the italics. Does the species occurs in Australia? Please only use the extent you used in Figure 1.
Figure 6: The legend is duplicated. C. punctatus is missing the italics. Please only use the extent you used in Figure 1
Figure 7: The legend is duplicated. C. punctatus is missing the italics. Please only use the extent you used in Figure 1
Figure 8: The legend is duplicated. C. punctatus is missing the italics. Please only use the extent you used in Figure 1
Figure 9: The legend is duplicated. C. punctatus is missing the italics. Please only use the extent you used in Figure 1. Please increase the letters A and B in this figure.
Figure 10: The legend is duplicated. C. punctatus is missing the italics. Please only use the extent you used in Figure 1. Please increase the letters A and B in this figure.
L300: The heading title is in a different format.
L302: the citation needs to come before the period sign. Please use the acronym used before for "Bluetongue virus"
L303: Culicoides is missing the italics.
L305: The citation needs to come before the period sign.
L310: Please use the acronym used before for "Bluetongue virus"
L311: Culicoides is missing the italics.

---

## Round 0.4 · Major Revisions

Dear Dr. Al-Malki,

After another review round where I checked your text, I could still raise several issues that require your attention. Although you mentioned your text passed through a paid editing/proofreading, it still seems a deep editing is needed, considering the issues I showed below. Please take special care regarding your text and demand a thorough edition from the company you paid for your review since it is not satisfactory yet. I must say that this will be my last warning. I need to see a near impecable text in your next version to accept it. Otherwise, I will not be as condescending as I was before.

L19: Please include the name of the descriptor of the species, the year of description, and some taxonomic information in the first mention of the species in the title, abstract, and main text, as indicated by Packer et al. 2018.
Packer, L., Monckton, S. K., Onuferko, T. M., & Ferrari, R. R. (2018). Validating taxonomic identifications in entomological research. Insect Conservation and Diversity, 11(1), 1–12.
L33: Please avoid using possessives in academic texts. It is too colloquial.
L50: Please use en dashes in numerical ranges.
L70: There is no need to use uppercase letter in "Epizootic"
L75: AUSV was used only here and can be cut.
L78: BTV
L84: "4ºC"
L88 and so on: Please note that you have two different citation formats after here. Before you used superscript citations and now you have citaions in the same level as the text. Please be consistent! Either use one or the other. Not both.
L98: Please avoid using possessives in academic texts. It is too colloquial.
L109: Please avoid using possessives in academic texts. It is too colloquial.
L113: Please avoid using possessives in academic texts. It is too colloquial.
L116: You introduced SDM in the previous sentence. So, please be consistent and use the acronym indicated.
L118: What algorithm? Please link your sentences better.
L122-123 and 125-126: The examples of Scientific names without italics continue after three rounds of reviews!
L132: "The occurrence data for C. punctatus"...
L139: Please avoid using possessives in academic texts. It is too colloquial.
L143: Please give a space between lines 142 and 143
L146-147: Please explain better the origin of the future varables
L152-153: Please use en dashes in numerical ranges.
L163: Please eplain BIOCLIM. It is not clear why you used BIOCLM and then Maxent. Please clarify. Why didn't you use Maxent since the beginning?
L167: Please avoid using possessives in academic texts. It is too colloquial.
L169-174: Please provide a more in-deepth description of Maxent. Please cite its original manuscripts and the latest one from 2017.
L179: Please avoid using possessives in academic texts. It is too colloquial. Another Scientific name without italics.
L183: Please avoid using possessives in academic texts. It is too colloquial.
L189: Please provide, at least, the name of the manufacturer of this software.
L196: Please avoid using possessives in academic texts. It is too colloquial.
L207: Please give a space between lines 207 and 208
L210: Please avoid using possessives in academic texts. It is too colloquial.
L212: Please use en dashes in numerical ranges.
L214: TSS was introduced before. There is no need to use it again. Use TSS
L216: Please use the minus symbol instead of the hyphen.
L222: Please give a space between lines 222 and 223
L231: Please give a space between lines 231 and 232
L259: italics
L250: Please give a space between lines 250 and 251
L296: Please give a space between lines 296 and 297
Figures: Please use the same color ramp in all figures.
L301: BTV
L306: BTV
L307: Please see that there is a period sign and the citation goes after it. It is too strange for me. Please be consistent regarding your citations. This happens throughout the entire text when your citations are not in superscript.
Please improve the discussion: Is Maxent a silver bullet? Does it solve all of our issues regarding species distribution? Are sampling biases affecting the obtained results? Should we be concerned with our results? What are other studies involving viruses and models that reached similar or different results? Regarding future models, please discuss and compare your results to other studies involving similar systems and scenarios.
L327: Please avoid using possessives in academic texts. It is too colloquial.
L329: "Figure". Also please note that AUC and TSS may be "excellent" and the produced models still be poor. How to deal with this issue?
L333: "Figure"
L336: Please include the name of the descriptor of the species, the year of description, and some taxonomic information in the first mention of the species in the title, abstract, and main text, as indicated by Packer et al. 2018.
Packer, L., Monckton, S. K., Onuferko, T. M., & Ferrari, R. R. (2018). Validating taxonomic identifications in entomological research. Insect Conservation and Diversity, 11(1), 1–12.
L338: "Figures 6-10"
L339: Italics
L343: "Figures"
L350: Please include the name of the descriptor of the species, the year of description, and some taxonomic information in the first mention of the species in the title, abstract, and main text, as indicated by Packer et al. 2018.
Packer, L., Monckton, S. K., Onuferko, T. M., & Ferrari, R. R. (2018). Validating taxonomic identifications in entomological research. Insect Conservation and Diversity, 11(1), 1–12.
L357: "C. punctatus"
L375: What kind of errors?
L376: What kinds of differences?
L377: What advantage?
L379: Why is it outperformed?
L379-380: Why? Please improve your argument.

---

## Round 0.5 · accepted · Accept

The authors have addressed most of the comments in the prior Editor's final correspondence. The paper reads quite well with a few minor language issues that don't detract substantially from the overall presentation. The reviews have been mainly positive throughout the process.